# PARP1 allows proper telomere replication through TRF1 poly (ADP-ribosyl)ation and helicase recruitment

C. Maresca[1,7], A. Dello Stritto[2,6,7], C. D'Angelo[1], E. Petti[1], A. Rizzo[1], E. Vertecchi[3], F. Berardinelli[4], L. Bonanni[4], A. Sgura[4], A. Antoccia[4], G. Graziani[5], A. Biroccio 🄳 [1✉] & E. Salvati 🄳 [3✉]

Telomeres are nucleoprotein structures at eukaryotic chromosome termini. Their stability is preserved by a six-protein complex named shelterin. Among these, TRF1 binds telomere duplex and assists DNA replication with mechanisms only partly clarified. Here we found that poly (ADP-ribose) polymerase 1 (PARP1) interacts and covalently PARylates TRF1 in S-phase modifying its DNA affinity. Therefore, genetic and pharmacological inhibition of PARP1 impairs the dynamic association of TRF1 and the bromodeoxyuridine incorporation at replicating telomeres. Inhibition of PARP1 also affects the recruitment of WRN and BLM helicases in TRF1 containing complexes during S-phase, triggering replication-dependent DNA-damage and telomere fragility. This work unveils an unprecedented role for PARP1 as a "surveillant" of telomere replication, which orchestrates protein dynamics at proceeding replication fork.

[1] Translational Oncology Research Unit, IRCCS Regina Elena National Cancer Institute, Rome, Italy. [2] Department of Biology and Biotechnology "Charles Darwin", Sapienza University of Rome, Rome, Italy. [3] Institute of Molecular Biology and Pathology, National Research Council, Rome, Italy. [4] Department of Biology, Roma Tre University, Rome, Italy. [5] Department of Systems Medicine, University of Rome Tor Vergata, Rome, Italy. [6] Present address: Institute of Molecular Genetics "Luigi Cavalli-Sforza", National Research Council, Via Abbiategrasso 207, Pavia, Italy. [7] These authors contributed equally: C. Maresca, A. Dello Stritto. ✉email: annamaria.biroccio@ifo.it; erica.salvati@cnr.it

Telomeres are nucleoprotein structures at eukaryotic chromosomes termini deputed to DNA end protection. They are non-genic regions consisting of species-specific GC rich repeats bound to a six-members specialized complex, called shelterin, which regulates telomere length homeostasis and prevents undesired recombination by repressing different pathways of DNA damage response[1,2]. Telomere duplication initiates from a single origin of replication, located at sub-telomeres, moving unidirectionally towards the chromosome end. To this end, proceeding replication forks must cope with the compaction of telomeric heterochromatin and the presence of secondary structures (t-loops and G-quadruplex). Thus, telomere replication requires the action of several enzymes including helicases, topoisomerases, and exonucleases[3].

Telomere Repeat Binding Factor 1 and 2 (TRF1-2) are members of the shelterin complex that directly bind to telomeric duplex, as homodimers, in a sequence-specific manner. Moreover, they interact and recruit other shelterins and chromatin remodeling enzymes to assist DNA replication and repair[4,5]. TRF2 has been shown to play a crucial role in difficult to replicate regions such as telomeres and pericentromeres, relieving topological stress during replication fork progression[6–8], and facilitating telomere replication initiation upon stress conditions[9]. Specifically, loss of TRF1 has been shown to slow down replication fork progression at telomeres, consequently causing telomere fragility[10,11]. This effect is partially explained by the fact that TRF1 can recruit Bloom (BLM) RecQ helicase to replicating chromatin[5]. Nevertheless, the mechanism of telomere fragility formation has not been completely clarified.

PARP1 is the most abundant protein at chromatin after histones. Poly(ADP-ribosyl)ation (PARylation) by PARP1 is involved in various cellular pathways, including DNA damage response, transcription and chromatin organization. The immediate and robust PAR synthesis, locally produced at damaged sites, modifies protein-protein and protein-DNA interactions and serves as a molecular scaffold for the subsequent recruitment of chromatin modulators and DNA repair proteins[12]. Consistently, PARP1 is necessary to activate different DNA repair pathways and its inhibition confers synthetic lethality to mutations of DNA repair genes (i.e., BRCA1/2, PALB)[13].

At telomeres, PARP1 is implicated in DNA damage repair through activation of the alternative Non-Homologous End Joining (alt-NHEJ) and Homologous Recombination (HR) pathways[14]. Moreover, PARP1 interacts with and covalently modifies TRF2[15]. Telomere specific PARPs (Tankyrase 1 and 2) are known to modify TRF1 and to regulate telomere elongation and sister chromatids separation during mitosis. PARP1 is also enriched at telomeric chromatin during G-quadruplex stabilization, to resolve replication-dependent damage[16,17].

Here we investigate the constitutive role of PARP1 at replicating telomeres, unveiling a role of this enzyme as a key modulator of protein dynamics at replicating telomeres.

## Results

**PARP1 and TRF1 interact in S-phase**. To investigate the interplay between PARP1 and TRF1 in telomere replication, we firstly examined if the two proteins could interact.

HeLa cells were synchronized by double thymidine blockade (Fig. 1a, scheme) and, 15 minutes before the second thymidine pulse, the cells were labeled with BrdU to mark replicating DNA. The distribution of cells in the different phases of cell cycle was measured by flow cytometry after Propidium iodide (PI) staining (Supplementary Fig. 1) and by biparametric analysis (Fig. 1b). Quantitative analysis showed that at the end of synchronization

(T0) most of the cells (about 60%) were at the $G_1/S$ boundary and progressed into $G_2/M$ after 4 hours (T4) from release (Fig. 1c). Biparametric analysis of PI- and BrdU-positive cells allowed us to distinguish the proliferative compartment in Early-S (ES = T0), Mid-S (MS = T2) and Late-S/$G_2$ (LS/$G_2$) (Fig. 1b, d). Flow cytometric analysis of asynchronous population (AS) was performed as internal control (Supplementary Fig. 1, Fig. 1c).

To evaluate TRF1-PARP1 interaction, co-immunoprecipitation (Co-IP) experiment was performed at different time points (Fig. 1e). Interestingly, even though PARP1/TRF1 interaction was detectable in AS, an increase in PARP1/TRF1 affinity was observed as cells progressed through S-phase, reaching a maximum level in MS, followed by a decrease in LS/$G_2$ cells. The results, confirmed in BJ EHLT transformed human primary fibroblasts, indicated that this interaction could be functional to telomere replication (Supplementary Fig. 2).

To visualize a direct interaction between PARP1 and TRF1 in-situ in intact cells, we performed a Proximity Ligation Assay (PLA), which reveals co-localization between proteins less than 40 nm far from each other, a distance at which two proteins are supposed to directly interact (controls and experimental set up are shown in Supplementary Fig. 3). PLA spots were detected in the nuclei of HeLa and analyzed by deconvolution microscopy (Fig. 1f, g). Signal quantification showed that the average number of spots/nuclei progressively increases from ES to MS cells, confirming the strongest affinity between the two proteins during replication (Fig. 1g).

Co-IP experiment performed at the same time points between PARP1 and the TRF1 homologue TRF2, showed a maximum PARP1/TRF2 affinity when the cells were in MS and LS/$G_2$ phase of cell cycle, indicating that the two proteins could act at different times of cell cycle (Supplementary Fig. 4A). Moreover, exposure of cells to replication dependent DNA damage (achieved with HU or high dose Aphidicolin treatment) failed to increase either PARP1/TRF1 or PARP1/TRF2 interactions, suggesting that activation of DNA damage was not involved in this process (Supplementary Fig. 4B).

**TRF1 is covalently PARylated by PARP1**. PARP1 synthetizes linear and branched PARs from NAD+ monomers, covalently linked to specific aminoacidic residues of PARP1 itself (homo-modification) or specific target proteins (hetero-modification). To ascertain if TRF1 was directly modified by PARP1 enzyme, an in-vitro hetero-modification assay was performed, in which recombinant TRF1 was added to PARP1 enzyme in presence of NAD+ and activated DNA. The protein mixture was resolved onto PAGE and PARs covalently bound to PARP1 and TRF1 were detected by SDS PAGE followed by western blot (WB) analysis with an anti-PAR specific antibody (Fig. 2a). Purified full-length recombinant TRF1 (Supplementary Fig. 5) was PARylated by PARP1 as shown by the appearance of a band at approximately 63 KD, in samples in which TRF1 was added (overlapping with the anti-TRF1 detected band shown in the lower panel), compared to PARP1 signal alone where a higher molecular weight signal is present, corresponding to auto-PARylated PARP1. PARylation was further increased by cleaved DNA which stimulates PARP1 catalytic activity. TRF1 covalent PARylation was also assessed by incorporation of biotinylated NAD+ in the Poly ADP-ribose polymers, in the hetero-modification assay. As shown in Fig. 2b, the NAD+ incorporation is detected both at >100 KD (PARP1) and at <63 KD (TRF1) when TRF1 is present, after biotin-NAD+ addition. The capacity of PARP1 to bind TRF1 by non-covalent interaction with polymers was exploited in a non-covalent PARylation assay. Here, recombinant full length TRF1 was immobilized onto a

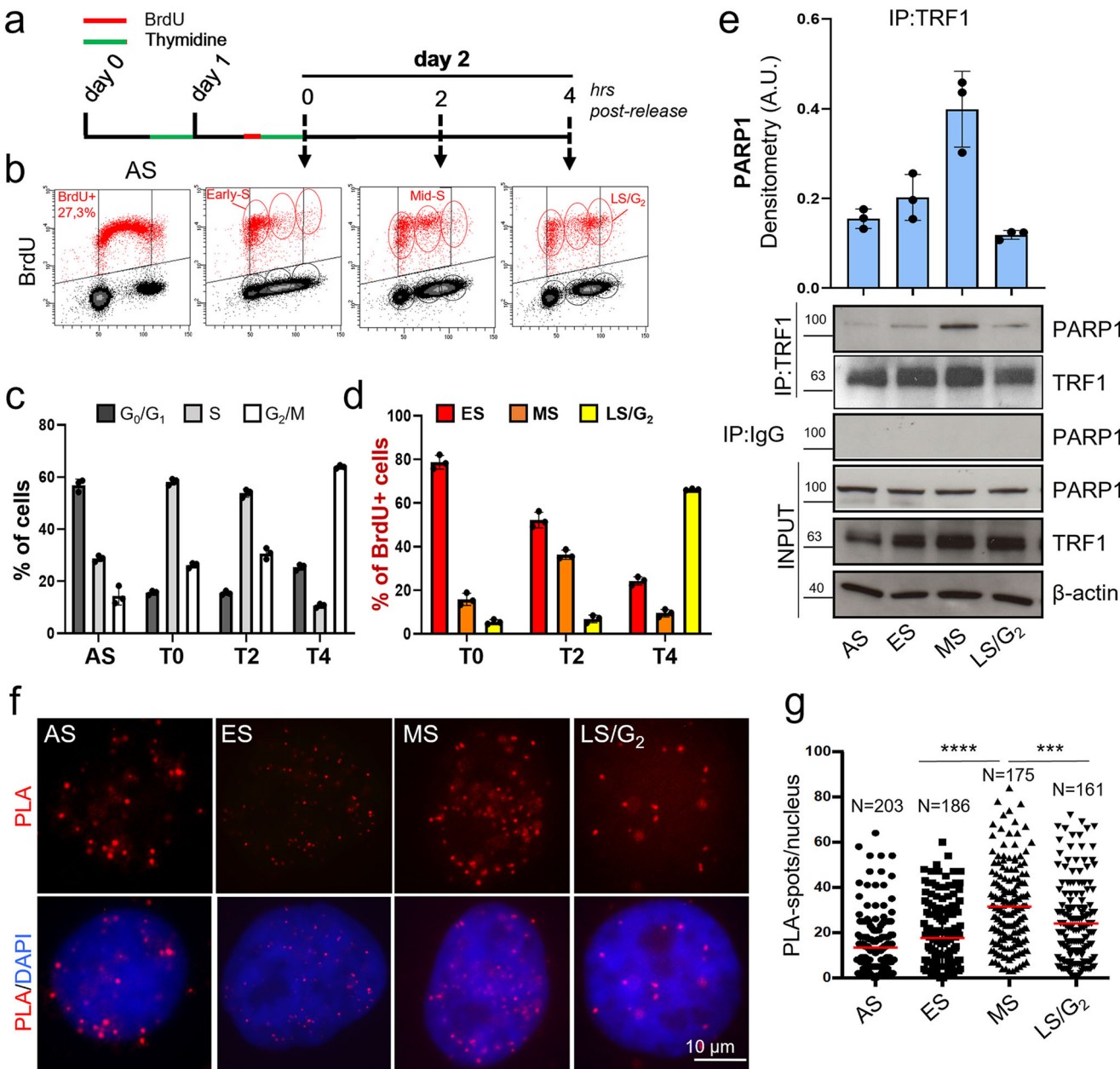

**Fig. 1 TRF1 and PARP1 interact during S-phase.** HeLa cells were synchronized in the early S phase by double thymidine block. A BrdU pulse was administered 15 minutes before the second thymidine block and then the cells were released and collected at the indicated time points. **a** At each endpoint, the DNA was denatured, incubated with anti-BrdU antibody and PI, and processed for flow cytometry. Bivariate distributions (dot plot) of BrdU content versus DNA content (**b**) were analyzed, boundaries between positive and negative samples have been indicated as a black line (one representative of three independent experiments is shown). Histograms report the quantification of the percentage of cells in the different cell cycle phases in the whole cell population (**c**) or in BrdU pulsed population (**d**, the mean of three independent experiments is shown, bars are SD), 1C. **e** Samples synchronized as in **a** underwent immunoprecipitation with an anti-TRF1 specific antibody or rabbit IgG as negative control followed by incubation with the indicated antibodies (β-actin was used as loading control, one representative of three independent experiments is shown). Western blot signals were quantified by densitometry and reported in histograms after normalization on IP-ed TRF1, and on input PARP1 and TRF1, background in IgG IP-ed samples was subtracted (the mean of three independent experiments is reported, bars are SD). **f** HeLa cells synchronized as described were fixed in formaldehyde and processed for PLA with anti-TRF1 and PARP1 antibodies. Signals were acquired by Leica Deconvolution fluorescence microscope at 63× magnification (representative images are shown). The number of signals/nuclei was scored by Image J software and reported in graph (**g**). For each column Mean (red bars) and numerosity (N) are indicated, two pulled independent experiments were plotted, P value was determined by unpaired two tailed t-student test, ***$P \leq 0.001$, ****$P \leq 0.0001$.

nitrocellulose membrane in parallel with the H1 histone (a known PARP1 substrate for both covalent and non-covalent PARylation) and incubated with in-vitro synthesized PARs, followed by anti-PAR detection. The dot-blot in Fig. 2c revealed that TRF1 is not able to bind polymers in non-covalent manner.

To confirm TRF1 PARylation in-vivo, HeLa cells were synchronized as before, and samples, collected at different time points, underwent anti-PAR immunoprecipitation and revealed with an anti-TRF1 antibody (Fig. 2d). For experimental specificity, cells were transfected with siTRF1 or scrambled

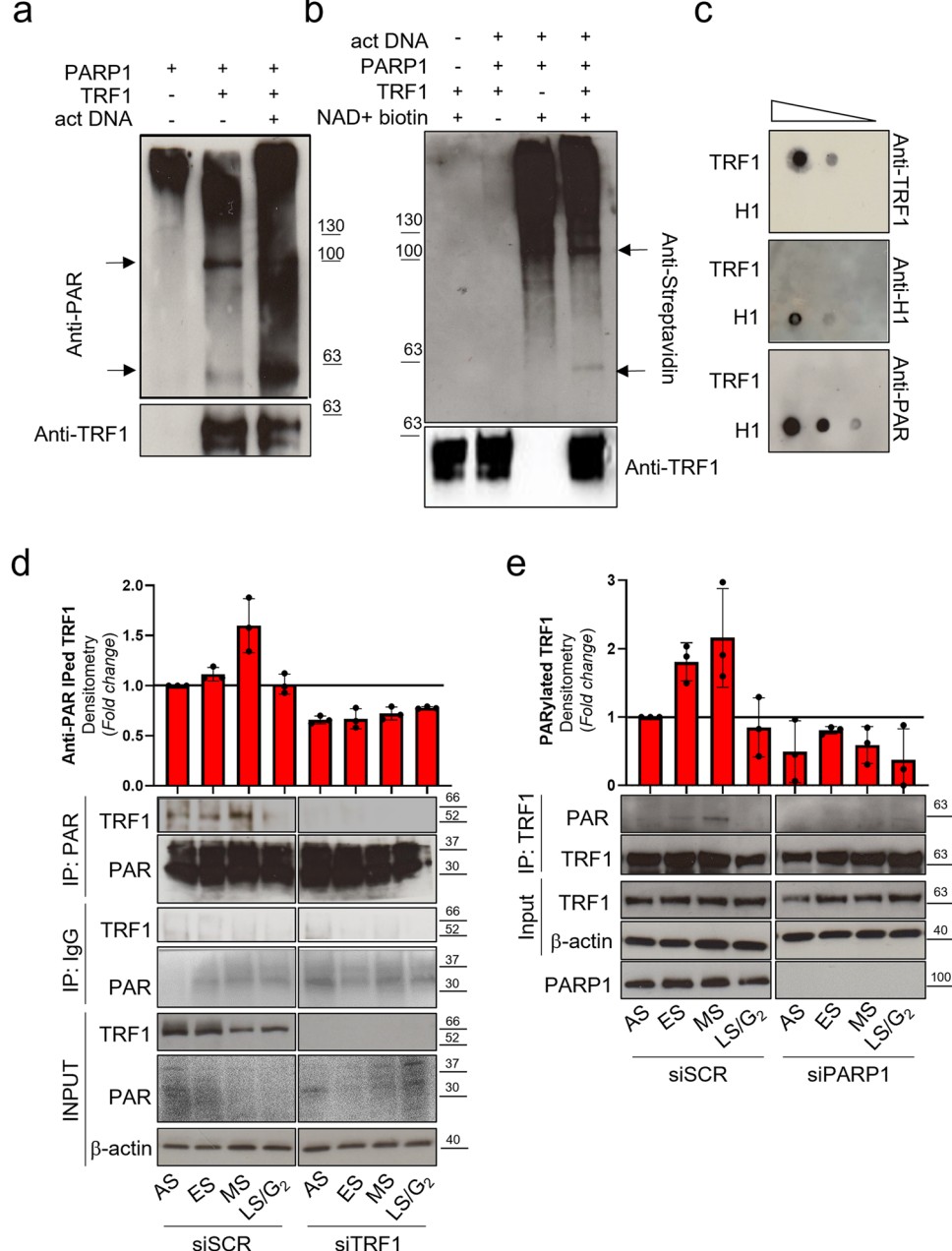

**Fig. 2 TRF1 is PARylated by PARP1 in-vitro and in-vivo. a** Covalent PARylation assay was performed by incubating a high activity purified PARP1 enzyme in the NAD+ containing PARylation buffer in absence or presence of recombinant His-tag full length TRF1, activated DNA was added to further stimulate the PARylation reaction. **b** The PARylation reaction was performed as above described in presence of biotynilated NAD+. In both **a** and **b** Protein mixtures were resolved on SDS-PAGE and incubated with an anti-PAR (**a**) or anti-Streptavidin (**b**) antibodies, to detect polymers, or Anti-TRF1. Signals were revealed by chemiluminescence. **c** Noncovalent PARylation assay: increasing quantities of recombinant full length TRF1 or H1 histone were spotted on nitrocellulose by dot blot, incubated with previously synthetized and purified PARs, and then incubated with anti-PAR antibody (to detect bound polymers). Pictures are representative of three independent experiments with similar results. **d, e** HeLa cells were transfected as indicated, then synchronized as previously described, collected and immunoprecipitated with the indicated antibodies or IgG as control. Input and immunoprecipitated samples were processed for Western blot analysis of the indicated antigens. Representative images of three independent experiments are shown. Western blot signals were quantified by densitometry, and the mean of three independent experiments is shown in the histograms, bars are SD.

sequence (siSCR). PAR immunoprecipitation efficiency was checked with the anti-PAR antibody showing an enrichment of PARylated proteins especially at low molecular weight (as expected since histones are heavily PARylated) (Fig. 2d). The results shown in Fig. 2d revealed that, consistently with the timing of TRF1/PARP1 interaction, TRF1 is Parylated by PARP1 in S-phase. The mirror experiment, where samples were immunoprecipitated with an anti-TRF1 and revealed with anti-

PAR antibody, confirmed the presence of Parylated TRF1 in S-phase (Fig. 2e). The absence of the TRF1 band in siTRF1 interfered and IgG immunoprecipitated samples evidenced the antibody specificity. The presence of an acceptor site for PARylation by Tankyrase 1 in the acidic N-terminal domain of TRF1 was previously described[18]. Then, we performed an in-vitro PARylation assay with a TRF1 deletion mutant lacking the acidic domain. As shown in Supplementary Fig. 6, the delta acidic

domain is still PARylated, indicating that the PARP1 PARylated domain is different from the Tankyrase 1 acceptor domain.

**TRF1 PARylation impacts on telomeric DNA replication**. PARylation is known to alter the chemical environment of target proteins modifying their interactions with other proteins and/or nucleic acids. It has been shown that TRF1 has a peculiar dynamic at telomeres during replication, detaching from chromatin during the replication fork passage[19]. Since PARP1 interacts with and PARylates TRF1 during S-phase, we wanted to ascertain if this interplay had a role in TRF1 dynamic and BrdU incorporation at replicating telomeres. To this aim, HeLa cells were interfered for PARP1 or a scrambled sequence and synchronized by double thymidine blockade (Supplementary Fig. 7). Then, 1 hr before sample collection, cells were exposed to BrdU incorporation as indicated in Fig. 3a. Samples collected at the indicated time points were analyzed by WB for PARP1 depletion efficiency (Supplementary Fig. 7A), ChIP against TRF1 to analyze its association to telomeric chromatin and by BromoIP assay to assess the replication fork passage (Fig. 3b, c). Analysis of control samples demonstrated that BrdU was incorporated during S (ES and MS) and $G_2$/M phases of cell cycle, consistently with an early and late telomere replication and processing (Fig. 3b, c, Supplementary Fig. 7B, C). The BrdU incorporation pattern was accompanied by a transient decrease in TRF1 association with telomeric chromatin (Fig. 3b, c). Notably, a marked decrease in the amount of BrdU incorporation associated with a lack of TRF1 dissociation in ES and MS from telomeric chromatin was observed in PARP1 depleted cells (Fig. 3b, c). The same results were confirmed by using the PARP1 pharmacological inhibitor olaparib (Supplementary Fig. 8A–D). Indeed, at a dose unable to trigger DNA damage response activation (Supplementary Fig. 8E), olaparib treatment induced an impairment of BrdU incorporation associated with an increase of TRF1 association to the telomeric chromatin. Our results suggest that PARylation of TRF1 by PARP1 can interfere with a proper telomere replication by modifying its affinity to DNA.

To verify this hypothesis the interaction affinity between unmodified or PARylated TRF1 with the telomeric duplex DNA was evaluated by Electro Mobility Shift Assay. The results reported in Fig. 3d show that unmodified TRF1 efficiently bound $^{32}$P-labeled telomeric duplex DNA, inducing a shift of the probe signal, while the binding efficiency was massively reduced when previously PARylated TRF1 was added to the reaction mix (Fig. 3d). As a control, PARP1 alone (which was present in the PARylation reaction) did not affect DNA migration.

Then, we investigated the role of PARP1 in the recruitment of helicases at replicating telomeres to solve secondary structures during the fork passage. Hela cells, synchronized as above and depleted for PARP1, were subjected to immunoprecipitation with an anti-TRF1 antibody to reveal the presence of helicases in the complex (Fig. 3e, f). In agreement with the literature TRF1 formed a complex with BLM[5]; interestingly, an unprecedented interaction with WRN was also observed in the samples (Fig. 3e, f). Of note, the interaction among TRF1 and the helicases mainly occurred in S-phase, and more importantly it was dependent on PARP1 (Fig. 3e, f). Indeed, ChIP analysis of WRN association to telomeric chromatin also showed a recruitment in S-phase that was completely abrogated in PARP1 interfered cells (Fig. 3g, h). To deeper characterize the dynamics of PARP1/TRF1/helicases at DNA, we performed the same immunoprecipitation in Fig. 3e in presence of EtBr. As shown in Supplementary Fig. 9A, TRF1/PARP1 interaction is still detectable in EtBr treated samples, but intercalation of EtBr into DNA abrogates part of the increase of affinity between TRF1 and

PARP1 observed in S-phase, especially in the MS, and the helicase recruitment. This is in line with the fact that TRF1 and PARP1 directly interact, as also confirmed by the pull-down experiment shown in Supplementary Fig. 9B. However, the S-phase specific TRF1/PARP1/helicases complexes seem to require DNA to form stably.

Given that the TRF1 paralog TRF2 is known to interact with RecQ helicases as well, and that TRF1 and TRF2 could interact, to further ascertain the specificity of TRF1 binding to helicases we analyzed the TRF2 affinity to helicases and PARP1 during S-phase upon TRF1 knock-down. As shown in Supplementary Fig. 10, TRF2 binds PARP1, WRN and BLM at each time point. Interestingly, WRN and BLM binding to TRF2 is neither modulated by cell cycle progression nor by TRF1 interference. More interestingly, PARP1 binding to TRF2 appears to increase in S-G2 phases (as also shown in Supplementary Fig. 4A), but this increase appears to be TRF1 dependent.

**PARP1 inhibition causes replication-dependent DNA damage and telomeric fragile sites**. Finally, we evaluated the cellular consequence of impairing replication fork progression at telomeres by siPARP1, by analyzing the cell cycle progression of BrdU-labeled cells at different times after BrdU pulse. As shown in Supplementary Fig. 11A, B, PARP1 interfered cells incorporated less BrdU at time 0. Moreover, the analysis of the BrdU-positive cells in the different phases of cell cycle evidences that cells depleted for PARP1 showed a delay in the progression through S to $G_2$: at 2.5 hours, a higher percentage of cells in S phase was observed in siPARP1 compared to siSCR sample ($50.8 \pm 1.7$ vs $39.1 \pm 0.7$), with a concomitant decrease in $G_2$ ($47.6 \pm 2.0$ vs $60.3 \pm 0.8$), and a similar trend was still detectable at 5 hours. The impairment of cell cycle progression is still evident at 24 h where a higher percentage of PARP1 interfered cells were in the $G_1$ phase of cell cycle ($43.5 \pm 0.4\%$ vs $29.3 \pm 0.1\%$), indicating a delay entering in a second duplication round.

With the aim of analyzing the occurrence of replication stress induced DNA damage, we monitored the activation of DNA damage response (DDR), revealed by the appearance of γH2AX foci at telomeric sequences in immunofluorescence/FISH single cell analysis at different time points after PARP1 or TRF1 down-regulation (via RNAi, Supplementary Fig. 12A), as a control of telomere replication perturbation (Fig. 4a–c). The single cell analysis showed a transient increase of the percentage of Telomere-dysfunction Induced Foci (TIFs)-positive cells (revealed as cells with more than 4 γH2AX/telomere probe colocalizations) and of TIFs mean number/nucleus in both TRF1 and PARP1 interfered samples, most of which were recovered at 72 h after transfection (Fig. 4b, c). In addition, in double interfered samples, the percentages of TIFs positive cells were comparable to the TRF1 single interfered ones, indicating that PARP1 and TRF1 acted in the same pathway (Fig. 4b, c). To confirm that the DNA damage induction was associated to replication stress, in the above conditions we measured the activation of RPA (replication protein A), which is phosphorylated by ATR and recruited at ssDNA during the resolution of the replication fork stall to protect the ssDNA from exonuclease attack[20]. The percentage of cells displaying co-localizing pRPA/telomeric signals (>4 co-localizing spots per cell) was scored, showing an increase upon both siPARP1 and siTRF1. Coherently with the above results, the double interference did not increase the percentage of positive cells (Supplementary Fig. 12).

To deepen the role of PARP1/TRF1 interaction in telomere replication, we evaluated the effect of a replication stress induction by low dose aphidicolin treatment. As shown in Fig. 5, low dose of aphidicolin treatment was able to induce an

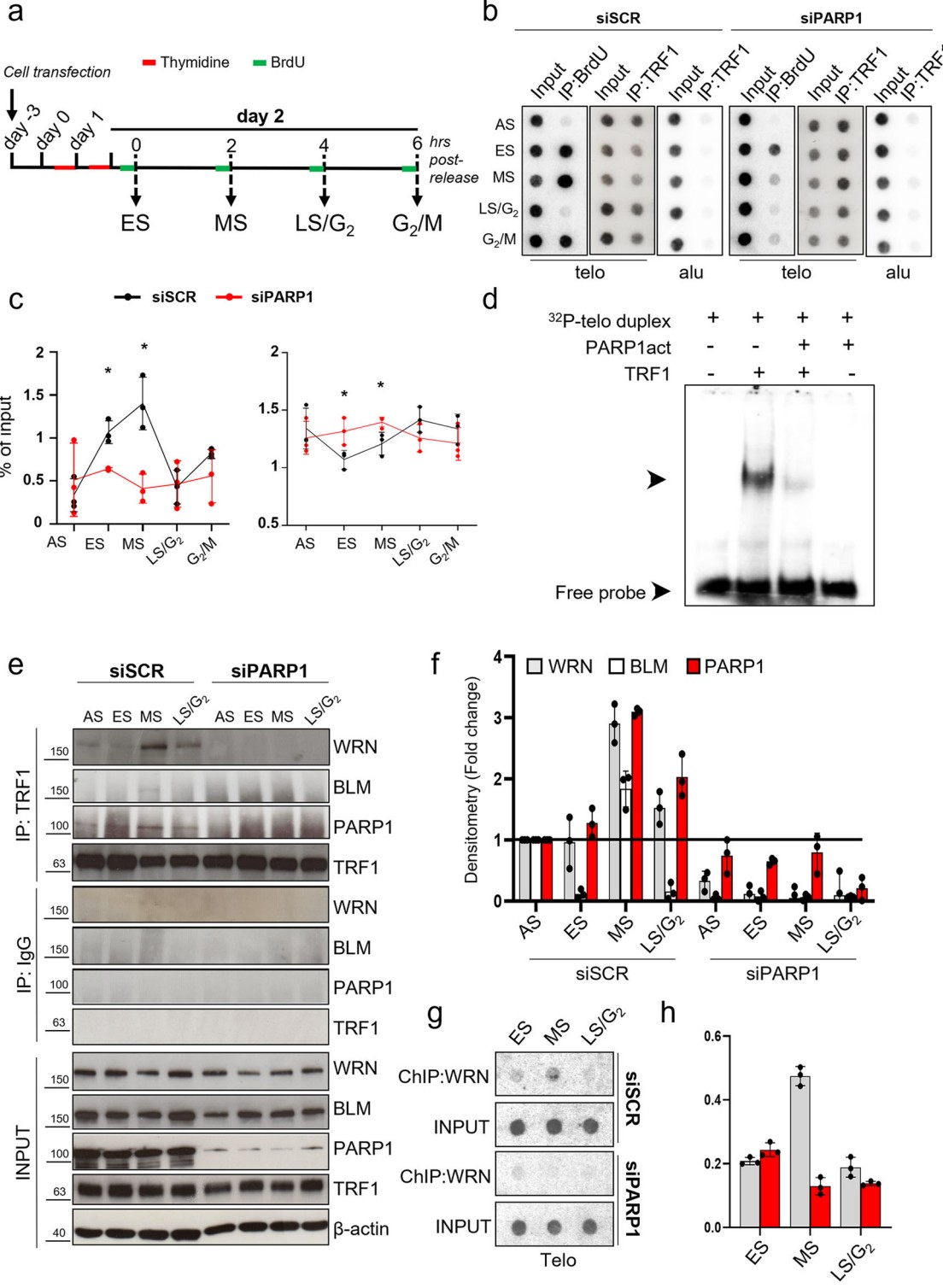

accumulation of cells in S-G$_2$M phases of cell cycle (Fig. 5a). In agreement with previous results showing an increase of PARP1/TRF1 interaction in S-phase, aphidicolin treatment also increased PARP1 immunoprecipitation by TRF1 and TRF1 PARylation (Fig. 5b). Coherently, aphidicolin treatment induced an enhanced percentage of TIFs positive cells at a basal level, and a further increase in presence of siPARP1 interference, confirming the role of PARP1 in resolving replication stress at telomeres. The mean number or TIFs/cell, scored among the TIFs positive cells, was slightly increased in basal condition but not further increased in siPARP1 cells (Fig. 5c–e).

Replication stress at telomeres is known to associate with a fragility phenotype, recognizable by the presence of a double telomeric spot at a single chromatid end in telo-FISH assay on metaphasic chromosomes. It has been recently proposed that telomere fragility could result as a secondary consequence of DNA repair of damaged replication forks by the homologous DNA recombination machinery. Then we exploited the

**Fig. 3 PARP1 inhibition perturbs DNA synthesis and TRF1 dynamics at telomeres in S-phase.** HeLa cells were transfected with siSCR and siPARP1 RNAs, synchronized by double thymidine block and exposed to BrdU incorporation one hour before each endpoint (**a**). Samples were processed for ChIP against TRF1 or BrdU IP. Immunoprecipitated chromatin was dot blotted and hybridized with a radiolabeled probe against telomere repeats or Alu repeats (**b**). Immunoprecipitated samples signals were quantified by densitometry, and then reported as the percentage of immunoprecipitated chromatin on each respective input, Alu signals are shown as control of antibody specificity in TRF1 ChIP (**c**). One representative of three independent experiments with similar results is shown in the pictures in **b**, graphs in **c** report the mean of three independent experiments, bars are SD. **d** EMSA assay, radiolabeled telomeric DNA duplex was incubated with unmodified or PARP1-PARylated TRF1 and run-on nondenaturing polyacrylamide gel. Signals were acquired at the Phosphoimager. One of three independent experiments with similar results is shown, arrows indicate labelled DNA shift. **e** HeLa cells interfered for PARP1 and synchronized as above reported were immunoprecipitated against TRF1 and incubated with the indicated antibodies. Densitometry of immunoprecipitated proteins, normalized on each respective input and on the level of immunoprecipitated TRF1 is reported in the histogram as fold change with respect to AS sample. The histogram shows the mean of three independent experiments, bars are SD (**f**). The indicated samples, synchronized as in **a**, underwent ChIP against WRN. Immunoprecipitated chromatin was dot blotted and hybridized with a radiolabeled probe against telomere repeats or Alu repeats (**g**) Signals from ChIPed chromatin in each sample was quantified by densitometry and reported as percentage of each relative input (**h**). One representative of three independent experiments with similar results is shown. Graphs report the mean of three independent experiments, bars are SD.

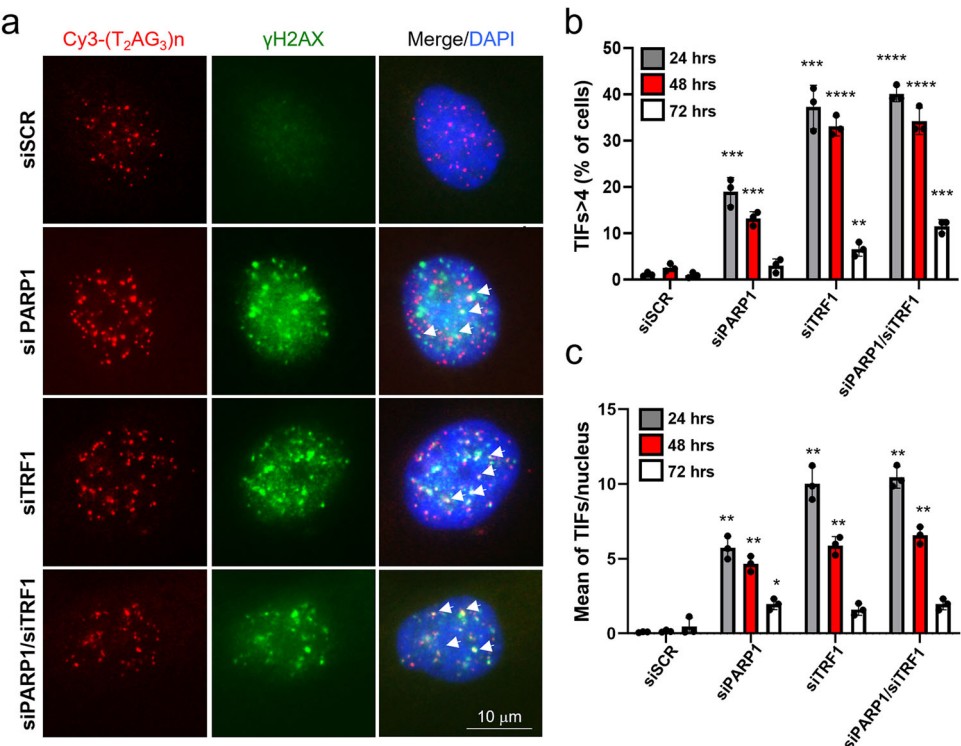

**Fig. 4 PARP1 inhibition causes transient DNA damage at telomeres. a** HeLa cells were transfected with siRNAs against siPARP1, siTRF1 or a scrambled sequence, alone or in the indicated combinations and against a scrambled sequence. Then samples were fixed at the specified endpoints after transfection and processed for IF-FISH against γH2AX and telomere repeats by a Cy3-Telo PNA probe and counterstained with DAPI. Signals were acquired by Leica Deconvolution fluorescence microscope (representative images at 63X magnification are shown), the percentage of TIFs positive cells (displaying >4 γH2AX/telomere co-localizations) and the mean number of TIFs/nucleus among γH2AX displaying nuclei were scored and reported in histograms in **b** and **c** respectively. The average of three independent experiments is shown, bars are SD. *P* value was determined by unpaired two tailed *t*-student test *$P \leq 0.05$, **$P \leq 0.01$, ***$P \leq 0.001$, ****$P \leq 0.0001$.

possibility that the transient telomeric damage observed upon PARP1 and/or TRF1 interference could be followed by the appearance of fragile telomeres. To this aim, cells were processed for FISH analysis on chromosomes spreads after 72 hours after siRNA transfection (efficiency of RNAi is shown in Supplementary Fig. 13A). Interestingly, PARP1 interference (Fig. 6a, b), as well as pharmacological inhibition (Supplementary Fig. 13B, C), were able to induce a significant increase of telomere fragile sites, comparable to the TRF1 interference alone, and the double TRF1/PARP1 knock-down displayed similar results compared to the PARP1 knock-down. Consistently with our data, neither TRF1 nor PARP1 or their combination resulted in telomere attrition (Supplementary Fig. 14A, B), or other kinds of telomere

aberrations not related to telomere replication (Supplementary Fig. 14C). Since the PARP1/TRF1 complex recruits WRN helicase, to investigate the contribution of this last in the resolution of replication stress, we also analyzed the effect of WRN depletion on telomere fragility. As shown in Fig. 6, WRN depletion induced telomere fragility as expected and already shown[21]. More interestingly, the double interference with TRF1 or PARP1 did not increase the incidence of telomere fragility confirming the epistatic relationship between PARP1, WRN and TRF1. Finally, since cells with longer telomeres can experience a higher replication stress, we tested the effect of PARP1 and TRF1 interference in Hela 1.3 clone which possesses 23 kb long telomeres (compared to parental Hela cell line which have 4 Kb

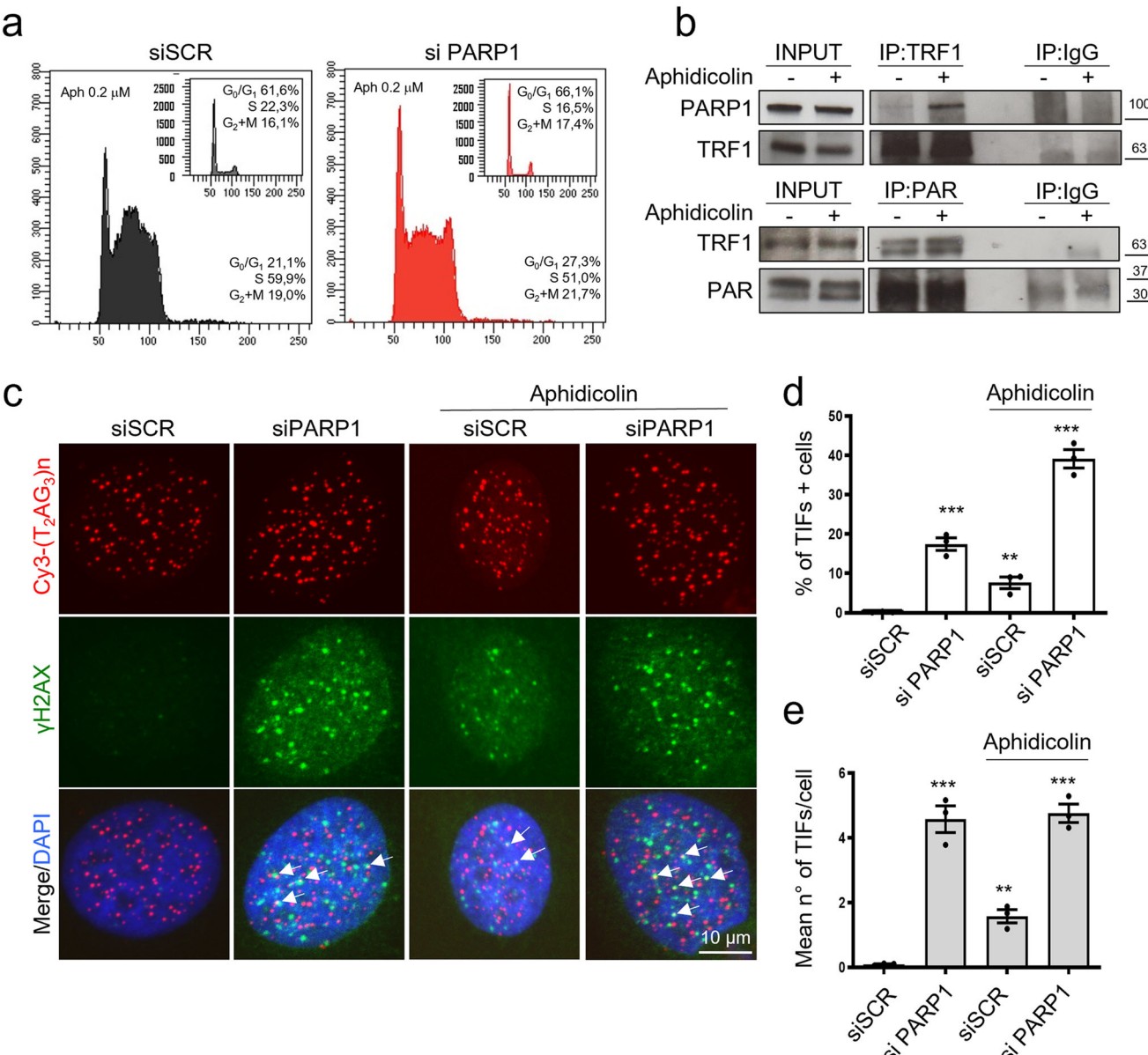

**Fig. 5 Aphidicolin induced replication stress increases TRF1/PARP1 interaction, TRF1 PARylation and TIFs induction upon PARP1 interference.** HeLa were treated with 0,2 µM aphidicolin for 20 h Then cells were PI stained and analyzed for cell cycle distribution by flowcytometry (**a**). Parallel samples were processed for IP with an anti-TRF1 followed by anti-PARP1 to measure TRF1/PARP binding, or with an anti-PAR followed by an anti TRF1 antibody to detect TRF1 PARylation (**b**). Hela cells interfered with siPARP1, or a SCR sequence were exposed to 0.2 µM aphidicolin for 20 h and then processed for IF-FISH to detect γH2AX and telomeric sequences. Representative images at 63X magnification were shown (**c**). The percentage of TIFs positive cells and the mean number of colocalization per nucleus among the γH2AX positive nuclei were scored and reported in histograms **d** and **e** respectively. The histograms report the mean of three independent experiments. Bars are SD. P value was determined by unpaired two tailed t-student test **$P \leq 0.01$, ***$P \leq 0.001$.

long telomeres). As shown in Supplementary Fig. 15, in this cell line, the occurrence of doublets/chromosome was slightly increased with respect to parental cells also at the basal level (0.09 vs 0.12 d/c). More interestingly, a stronger increase was observed, with respect to parental cells, upon siPARP1 (1.3 vs 1.9 d/c), siTRF1 (1.3 vs 2.6 d/c), and double interference (1.5 vs 2.0 d/c) (Fig. 6).

## Discussion

PARP1 is canonically activated at DNA lesions where it synthetizes PAR chains on itself and on specific acceptor proteins, modifying the molecular environment around the damaged site and facilitating repair actions. In this paper, we highlighted functions of PARP1 at telomeres during DNA synthesis that are

required for proper DNA replication. As first, we assessed a physiological interaction between endogenous PARP1 and TRF1, that is strongly increased in S-phase in the absence of DNA damage induction (Fig. 1 and Supplementary Figs. 1–4). Then, we discovered that TRF1 is a substrate for PARP1-dependent covalent PARylation both in in-vitro assay and in cells (Fig. 2). PARylation in-vivo is dependent on PARP1, since PARP1 interference can abrogate it. In addition, the timing of PARP1/TRF1 interaction and TRF1 PARylation, nicely correlated, starting in the ES, rising in the MS, and returning to basal level in LS/$G_2$. This evidence, together with the well-established role of TRF1 in telomere replication[5,10,11,22], suggested us to deepen the contribution of PARP1 and PARylation during telomere replication by Bromo-ChIP experiments.

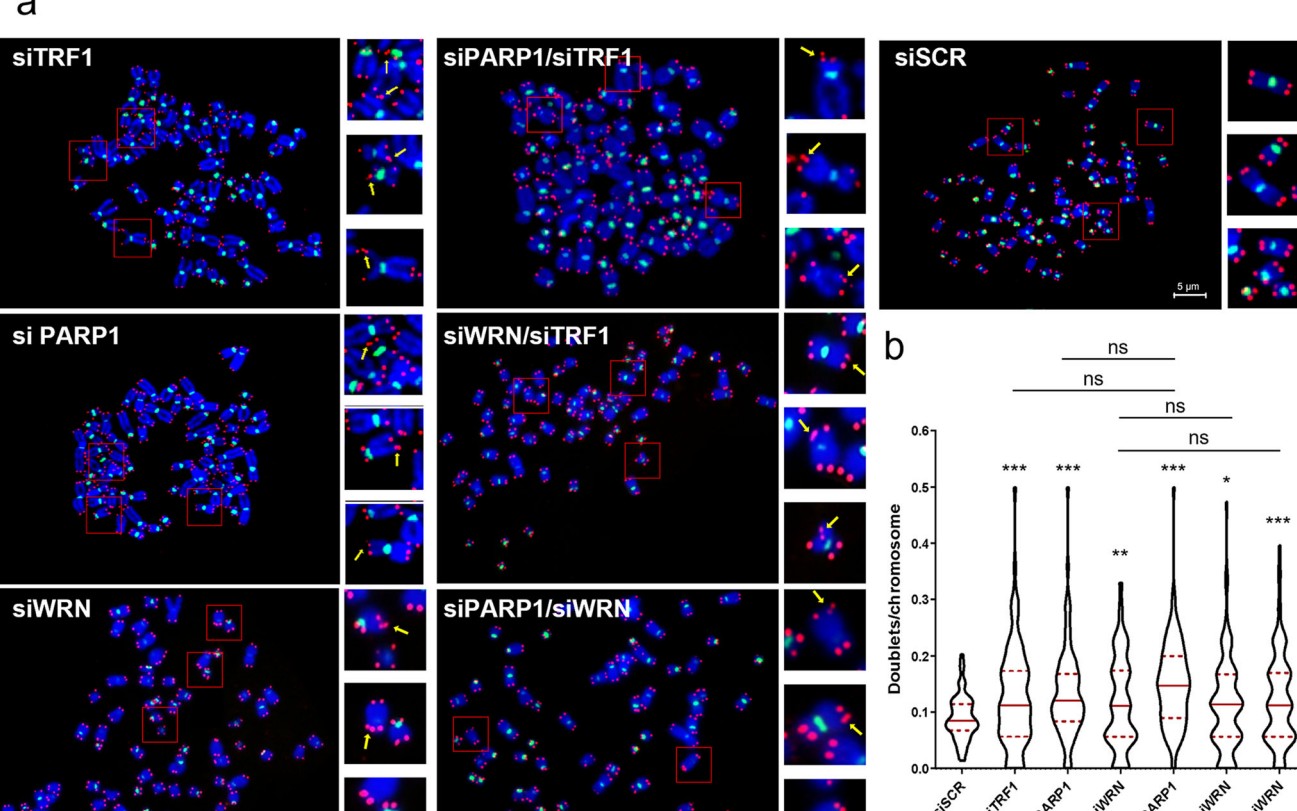

**Fig. 6 PARP1 inhibition induces telomere fragility. a** HeLa were transfected as indicated and after 72 hours metaphases were collected and processed for FISH analysis with pantelomeric/pancentromeric staining and counterstained with DAPI. Representative images at 100× magnification are shown. **b** Telomere doublets were scored and reported in graphs as the number of doublets/chromosomes. Three pulled independent experiments were plotted, 100 metaphases per sample were scored, red bars are means, P value was determined by unpaired two tailed t-student test. **P ≤ 0.01, ****P ≤ 0.0001.

The role of PARylation in modulating protein-DNA affinity is a common notion across the literature, mainly since PARylation adds negative charges to PARP1 itself and to its targets. PARylation has been specifically involved in the eviction of histones from damaged DNA sites and in the displacement of DNA repair proteins[23,24]. Based on this concept, we hypothesized that TRF1 PARylation decreased TRF1 binding to DNA duplex, facilitating the access of the replisome. In agreement with our hypothesis, EMSA assay revealed that PARylation by PARP1 impaired TRF1 binding to telomeric duplex, and, coherent with this, Bromo-ChIP experiments showed a decrease of TRF1 binding to telomeric chromatin in ES (Fig. 3a–d).

Replication fork progression at telomeres has been shown to require RecQ helicases activity to unwind secondary structures[4,22]. BLM and WRN RecQ helicases are covalent and non-covalent PARs binders interacting with PARP1[25]. Here we found that TRF1 co-immunoprecipitated both BLM and WRN helicases in S-phase, and, more importantly, PARP1 was shown to be required to maintain the TRF1/helicases complexes and to recruit WRN at telomeric DNA (Fig. 3e–g). The specificity of this interaction was also assessed by analyzing the TRF2 affinity for the three proteins in presence of TRF1 interference. TRF2 is known to interact with helicases and PARP1, however, helicase binding was found unaffected by cell cycle progression, while PARP1 affinity increase in S-phase was found to be mediated by TRF1, which further support a prevalent role of TRF1 in telomere replication with respect to TRF2 (Supplementary Figs. 4 and 10). To further dissect the TRF1/PARP1 interplay in S-phase, we

analyzed the affinity of the two proteins, and the helicase recruitment, in presence of EtBr, which disrupts DNA-protein interactions. We interestingly observed that the TRF1/PARP1 interaction was only reduced by the treatment, but not completely abrogated (Supplementary Fig. 9A), confirming that there is a direct interaction, that does not require DNA, as also shown by the pull-down experiment (Supplementary Fig. 9B). However, the increase in TRF1/PARP1 affinity and the helicase recruitment observed especially in the MS, were abrogated by the treatment. This interesting observation prompted us to formulate the hypothesis that the TRF1/PARP1 interplay has a dual effect. We propose a model in which PARP1 assists telomere replication: i) PARP1 PARylates TRF1 through a direct interaction in order to dissociate it from telomere duplex allowing replication fork opening; ii) TRF1/PARP1 complex is stabilized and recruits WRN and BLM helicases on the G-rich strand to resolve secondary structures, allowing the replication fork progression (Fig. 7). In agreement with this, PARP1 depletion, not only disrupts TRF1/helicase binding, but also impairs WRN recruitment at telomeric DNA in MS. The dynamic of this two-step model is still to be investigated at a biochemical level, since for instance we do not know how PARs addiction could modulate PARP1/TRF1 binding, or helicase recruitment. Anyway, this model fits with the phenotypical effects observed in the cells: in PARP1 interfered cells we observed a slight delay in the progression through S-phase, that is coherent with a perturbation of DNA synthesis. In addition, PARP1 interference induced a replication dependent DNA damage at telomeres, with an extent comparable to TRF1

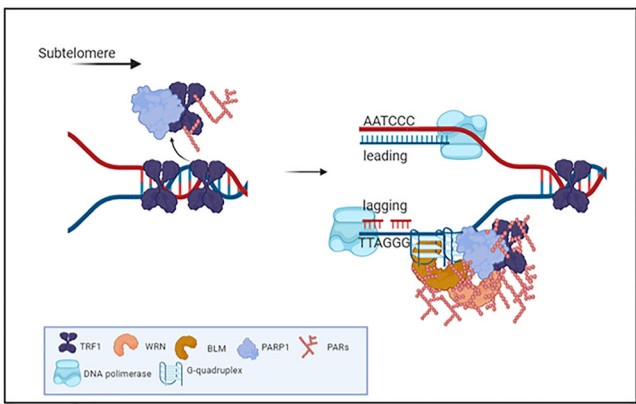

**Fig. 7 Model for TRF1/PARP1/RecQ helicases interplay during telomere replication.** In the ES, PARP1 binds to and PARylates TRF1 decreasing its affinity for DNA duplex, that is functional for fork progression. In MS PARP1/TRF1 complexes are stabilized on single stranded DNA and recruit BLM and WRN that resolve secondary structures forming at the G-rich lagging strand. Created in BioRender.com.

interference and, more importantly, the double interference did not increase the percentage of damaged cells, supporting the conclusion that TRF1 and PARP1 acted in the same pathway (Fig. 4). The DNA damage activation in consequence to replication stress was confirmed by the phosphorylation of RPA in a similar percentage of cells (Supplementary Fig. 12). Moreover, the increase of replication stress upon low dose aphidicolin treatment strengthened the TRF1/PARP1 binding, TRF1 PARylation and increased the percentage of cells with damaged telomeres, confirming the role of the TRF1/PARP1 axe in the resolution of replication stress at telomeres (Fig. 5).

It has been recently proposed that telomere fragility, a phenotype associated to replication stress at telomeres, could result from rearrangements occurring during replication fork stalling resolution by Homologous Recombination machinery[22]. In line with this, we found that siPARP1 and siTRF1 induced damage was recovered after 72 hours of transfection (Fig. 4). At the same time point, the metaphase telo-FISH analysis revealed the appearance of telomere fragility suggesting that DNA damage recovery involved recombination processes leading to telomere doublets. Both RNA interference and pharmacological inhibition of PARP1, were able to trigger telomere fragility, and the double interference with siTRF1 confirmed an epistatic relation between the two proteins. More interestingly, epistasis was also confirmed with WRN interference (Fig. 6). In agreement with the role of TRF1/PARP1 interaction in the assistance to telomere replication, cells with very long telomeres, which encounter major difficulty during telomere replication, also show a higher sensitivity to depletion of TRF1, PARP1 or both compared to parental cells (Supplementary Fig. 15).

Other authors reported that a stable PARP knock-out triggers loss of telomere repeats and DDR induction in colon cancer cells[26]. Here we did not observe any reduction of telomere length or increase of other telomeric defects, but this could be explained by the fact that we analyzed cells in the first round of duplication after a transient knock-down.

Taken together, all the data in the present work support a specific role for PARP1 in the replication of telomeres that should be considered also as a potential side-effect of PARP1 targeting treatments, that could generate telomere instability.

## Methods
**Cell cultures, transfections, and treatments**. Human cervical cancer cells (HeLa), BJ human fibroblasts and human osteosarcoma cells (U2OS) were purchased from ATCC repository. Hela 1.3 cells are a gift from Titia de Lange[27]. BJ

EHLT (hTERT immortalized, SV40 LT antigen transformed) were obtained by retroviral transduction of BJ cells with hTERT (Addgene plasmid #1773) and Large T SV40 antigen (Addgene plasmid # 21826). Cells were maintained in Dulbecco Modified Eagle Medium supplemented with 10% fetal calf serum, 2 mM L-gluta-mine, and antibiotics. In synchronization experiments, cells were seeded at 40% confluence and exposed to 2 mM thymidine (T-1895, SIGMA) for 16 h (I block), followed by 8 h release in fresh medium, and again exposed to 2 mM thymidine for additional 16 h (II block). Then cells were released in fresh medium and collected by trypsinization at different time points for further analysis. RNAi was performed by transfecting cells 2 days before synchronization at 20% confluence with 5 nM siRNA (scrambled sequence, two different sequences against PARP1, WRN and TRF1: PARP1 siRNA Origene SR300098B/C, TRF1 siRNA Origene SR322000B/C, SCR siRNA Origene SR30004, WRN siRNA ORIGENE SR322215B/C and POLYPLUS INTERFERIN #409-10 as Transfection reagent).

**Flow cytometry**. Cell cycle analysis was performed by flow cytometry (Becton-Dickinson) after cellular staining with propidium iodide (PI)[28,29]. After culturing and treatment, cells were harvested, washed with PBS twice, fixed in 70% ethanol at 4 °C overnight. Then, cells were washed with PBS twice, stained with PI at a final concentration of 50 μg/mL and RNase at a final concentration of 75 kU/mL, incubated for 30 min, then analyzed by FACSCalibur and FACSCelesta (BD Biosciences, San Jose, CA, USA). Progression of cells through the cell cycle phases was analyzed by simultaneous flow cytometric measurements of DNA and 5-bromo-2′-deoxyuridine (BrdU) contents of cells, as previously described[16]. Briefly, cells were pulsed with BrdU (Sigma Aldrich) at a final concentration of 20 μM for 15 min, and after the appropriate intervals in BrdU-free medium (from 2.5 to 24 h) the DNA was denatured. Cells were then incubated with 20 μl of the mouse Mab-BrdU (347580 Pure BD) for 1 hr at room temperature, and BrdU-labeled cells were detected using goat anti-Mouse Fab′2 Alexa Fluor 488 (Cell Signaling). The cells were counterstained with PI, acquired and analyzed with BD FACS Diva Software. During data acquisition a threshold trigger and FSC/SSC parameters has been used to gate out cellular debris. Cell clumps or doublets have been eliminated by using Doublet Discrimination Module (DDM) of FL3-A versus FL3-W.

**Immunoprecipitation and western blot**. Cells treated as above were collected and lysed in nuclei isolation buffer (10 mM Hepes pH 7.5, 10 mM KCl, 0.1 EDTA, 0.1 mM EGTA, 0.1 mM DTT, protease and phosphatase inhibitors). Nuclei were isolated by centrifugation and lysed in high salt RIPA buffer (50 mM Tris-HCl ph 7.4, 330 mM NaCl, NP-40 1%, DOC 0.25%, protease, and phosphatase inhibitors). For immunoprecipitation, 500 μg of proteins were incubated with 4 μg of goat IgG, anti-TRF2 (Mouse Mab 05521 Millipore), anti-TRF1 antibody (Goat Pab sc-1977, SantaCruz; Mouse Mab MA5-31596, Invitrogen), or anti-PAR (Mouse Mab 10H ALX-804220, Alexis; Rabbit Mab 83732 S, Cell Signaling) recovered with Protein-G and A dynabeads (10004D and 10002D, Invitrogen), run on PAGE together with input sample (1:20 of amount of immunoprecipitated proteins) and blotted with anti-PARP1 (Mouse Mab 551025 BD Pharmigen), anti-TRF1 (Rabbit Pab sc-6165, Santa Cruz), anti-PAR (mouse Mab 10H ALX-804220, Alexis), anti-WRN antibody (Rabbit Pab A300-239A, Bethyl Lab) or anti-BLM antibody (Rabbit Pab ab2179 Abcam; Mouse Mab sc-365753, Santa Cruz).; β-actin was used as a loading control (mouse Mab Sigma A2228).

**Protein expression and purification**. His-tagged human wild-type (wt) and delta acidic TRF1 were expressed in Escherichia coli Rosetta (DE3) competent cells by using pTrc-HisB vectors (a kind gift of Prof. Eric Gilson, University of Nice-Sophia Antipolis, Nice). TRF1 variants expression were induced at an OD 600 of 0.3–0.4 with 1 mM IPTG, followed by an incubation for 4 h at 37 °C. After centrifugation, cells were resuspended in lysis buffer [50 mM sodium phosphate pH 7.2, 300 mM NaCl, 10 mM imidazole, 1 mg/ml lysozyme, PMSF]. Cells were sonicated and the insoluble fraction was removed by centrifugation at 15,000 g for 30 min. The soluble fraction was loaded on 1.5 ml of HisPur™ Ni-NTA Resin (Thermo Fisher Scientific Inc.) and incubated 1 hr at 4 °C on rotation. Elution was performed with 250 mM imidazole in a buffer consisting of 50 mM sodium phosphate pH 7.2, 300 mM NaCl. Elution fraction was run on PAGE and quantified by Coomassie staining.

**Pull down assay**. Pull down assay was performed as previously reported[30] His-tagged human wild-type (wt) TRF1 was obtained as above described without elution from the HisPur™ Ni-NTA Resin (Thermo Fisher Scientific Inc.). Recombinant GST was expressed in bacteria and purified with Gluthation-Sepharose beads (Amersham) as reported[30]. Briefly, Hela cells were extracted in RIPA buffer with 330 mM NaCl. Recombinant proteins (8 pmol each sample) were incubated with 500 μg of cell lysate diluted in 1 ml of pull-down buffer (20 mM Tris–HCl pH 8, 100 mM KCl, 1 mM EDTA and 0.2% Triton) at 4 °C o.n. After incubation, beads were washed, resuspended in leamlli buffer and pulled down proteins were resolved in SDS PAGE.

**Heteromodification of HIS-hTRF1 isoforms by PARP1**. For the analysis of TRF1 PARylation by PARP-1[31], 160 ng of wt or delta acidic hTRF1/sample were incu-bated in the reaction buffer containing 5 units of hPARP-1 (High Specific Activity, Trevigen), 2.5 μg DNase I-activated calf thymus DNA, 200 mM NAD+, Tris-HCl

pH 8, 10 mM MgCl$_2$ and 2 mM dithiothreitol. After 30 min of incubation at 25 °C, the reaction was stopped by the addition of a Laemmli sample buffer and samples were analyzed by gel electrophoresis on 8% SDS-PAGE and Western blot. PARylated PARP1, hTRF1 or delta acidic hTRF1 were detected using anti-PAR monoclonal antibody (Trevigen) and input TRF1 was revealed with anti-His (anti 6-His Rabbit Pab Sigma Aldrich) or anti TRF1 antibody (rabbit Pab sc-6165, Santa Cruz). For the detection of biotin-labeled PARylated proteins the same assay was conducted in presence of biotin-NAD + (Sigma Aldrich) followed by SDS-PAGE and western blot detection with anti-streptavidin HRP antibody (Molecular Probes).

**Synthesis of PAR and non-covalent PAR binding**. Synthesis of PAR was performed as previously reported[20]. Briefly, 50 units of purified human PARP-1 (High Specific Activity hPARP-1, Trevigen) were incubated in a mixture containing 100 mM Tris-HCl pH 8, 10 mM MgCl$_2$, 2 mM dithiothreitol, 2.5 μg of DNase I-activated calf thymus DNA (Trevigen) and 200 mM NAD+ (Sigma-Aldrich) for 45 min at 30 °C. The reaction was stopped by adding ice-cold trichloroacetic acid (TCA) to a final concentration of 20% (w/v). PARs were detached from proteins by incubation in 50 mM NaOH and 10 mM EDTA for 1 hr at 60 °C. After adjustment of pH to 7.5, PAR were purified by phenol/chloroform extraction as described[20].

For the study of non-covalent interaction of PAR with TRF-1, graded concentrations of purified His-hTRF1 protein were immobilized directly by slot-blotting on nitrocellulose membranes. Histone H1 (Millipore) was used as positive control in the PAR binding assay. Subsequently, filters were incubated with PAR diluted in TBS-T (10 mM Tris-HCl pH 8.0, 150 mM NaCl, 0.1% Tween 20) for 1 hr at room temperature. After high-stringency salt washes, protein bound PAR were detected using the anti-PAR monoclonal antibody (mouse Mab ALX-804220).

**ChIP and BrdU-ChIP**. ChIP was performed after double thymidine blockade and PARP1 and TRF1 RNAi. Olaparib 2 μM (AZD2281 Selleckchem) and NU1025 200 μM (Sigma Aldrich) were given to cells during release from cell cycle blockade. Cells were collected every 2 hours post-release after addition of formaldehyde (1%) directly to culture medium for 10 min at R.T. and sonicated chromatin (80 μg/sample) was immunoprecipitated (IP) overnight at 4 °C with 4 μg of the anti-TRF1 antibody (goat Pab sc-1977, Santa Cruz) or the anti-WRN antibody (Rabbit Pab NB100-471, Novus Biologicals). Crosslink was then reversed with NaCl 5 M and DNA was extracted with phenol-chloroform method. Brdu-ChIP was performed after addition of 20 μM BrdU (5-bromo-2′-deoxyuridine, Sigma Aldrich) directly to HeLa culture medium for 1 h, then cells were collected and 60 μg of sonicated chromatin was incubated overnight at 4 °C with 20 μl of the anti-BrdU antibody (347580 Pure BD). Then, IP was performed as described above. After precipitation with each antibody, the precipitants were blotted onto Hybond-N membrane (Amersham), and telomeric repeat sequences were detected with a Telo probe (TTAGGG). A nonspecific probe (Alu) was also used. To verify that an equivalent amount of chromatin was used in the immunoprecipitated, samples representing the 0.1%, 0.01%, and 0.001% of the total chromatin (input) were included in the dot blot. The filter was exposed to a PhosphorImager screen (Bio-Rad), and the signals were measured using ImageQuant software (Quality One; Bio-Rad).

**Electro Mobility Shift Assay (EMSA)**. Telomeric duplex DNA 5′-GGGTTAGGG TTAGGGTTAGGGTTAGGGTTAGGGTTAGGGTTAGGGCCCCTC-3′ and antisense (5′-GAGGGGCCCTAACCCTAACCCTAACCCTAACCCTAACCCTA ACCCTAACCC-3′ was end-labeled with [γ-$^{32}$P] ATP (Amersham Biosciences) and T4-polynucleotide kinase (New England BioLabs) and purified from free nucleotides through G25 spin columns (GE Healthcare). Binding was carried out by incubating 0.5 ng of labeled DNA with 1 μg of unmodified or PARP1 covalently PARylated TRF1 (as above described) in 15 μl of a reaction mix of 20-mM Hepes (pH 7.9), 100-mM NaCl, 50-mM KCl, 1-mM MgCl$_2$, 0.1-mM ethylenediamine-tetraacetic acid (EDTA), 1 mM DTT, 5% (v/v) glycerol, 0.5 mg/ml of BSA and 0.1% (v/v) NP-40. Samples were incubated at 4 °C for 90 min and then run on native 4.5% polyacrylamide gels. Gels were dried and exposed to PhosphorImager screens and acquired using ImageQuant (Bio-Rad), and the signals were measured using ImageQuant software (Quality One; Bio-Rad).

**PLA, IF-FISH and FISH in metaphase**. For Proximity Ligation Assay (PLA) staining, HeLa cells, synchronized as above described, were fixed in 2% formaldehyde and permeabilized in 0.25% Triton X-100 in PBS for 5 min at room temperature at each endpoint. Then, samples were processed for immunolabeling with anti-TRF1 (rabbit Pab sc-6165, Santa Cruz) and anti-PARP1 (Mouse Mab ALX-804-211-R050, Enzo Life science) antibodies. PLA was performed by using the DUOLINK ® In situ detection reagents Red (Sigma-Aldrich) following the manufacturer's instructions. For IF-FISH staining, cells, fixed and permeabilized as indicated above, were immunostained with mouse anti-phospho-Histone H2AX (Ser139) (clone JBW301, Merk Millipore) or anti p-S4/S8 RPA (Rabbit Pab Bethyl A300-245A) monoclonal antibodies followed by the by the anti-mouse IgG Alexa fluor 488 or anti-rabbit IgG Alexa fluor 555 secondary antibodies (Cell Signaling). Then samples were re-fixed in 2% formaldehyde, dehydrated with ethanol series (70, 90, 100%), air dried, co-denaturated for 3 min at 80 °C with a Cy3-labeled PNA probe, specific for telomere sequences (TelC-Cy3, Panagene, Daejon, South Korea), and incubated for 2 h in a humidified chamber at room temperature in the dark.

After hybridization, slides were washed with 70% formamide, 10 mM TrisHCl pH7.2, BSA 0.1%, and then in TBS/Tween 0.08%, dehydrated with ethanol series, and finally counterstained with DAPI (0.5 μg/ml, Sigma-Aldrich) and mounting medium (Gelvatol Moviol, Sigma Aldrich). Images were captured at 63× magnification with a Leica DMIRE deconvolution microscope equipped with a Leica DFC 350FX camera and elaborated by a Leica LAS X software (Leica, Solms, Germany). This system permits to focus single planes inside the cell generating 3D high-resolution images. For telomere doublets analysis, chromosome spreads were obtained following 4 h incubation in colchicine 5 μM (Sigma-Aldrich) and pre-pared following standard procedure consisting of treatment with a hypotonic solution (75 mM KCl) for 20 min at 37 °C, followed by fixation in freshly prepared Carnoy solution (3:1 v/v methanol/acetic acid). Cells were then dropped onto slides, air dried, and utilized for cytogenetic analysis. Staining of centromeres and telomeres was performed as previously described[32] using the TelC-Cy3 PNA probe, and an Alexa488-labeled PNA probe specific for the human alphoid DNA sequence to mark centromeres (Cent-Alexa488) (both from Panagene, Daejon, South Korea). Metaphase images were captured using an Axio Imager M1 microscope (Zeiss, Jena, Germany) and the ISIS software (Metasystems, Milano, Italy). A total of 100 metaphases were analyzed for each sample in, at least, three independent experiments. Telomere length was calculated as the ratio between the relative fluorescence intensity of each telomere signal (T) and the relative fluorescence intensity of the centromere of chromosome 2 (C) and expressed as percentage (T/C %)[33].

**Statistics and reproducibility**. Experiments were replicated three times, with few exceptions as specified in the figure legends, and the data were expressed as means ± standard deviation (SD). GraphPad Prism 6 was used for the statistical analysis and the differences between groups were analyzed by the unpaired Student $T$ test. Differences were considered statistically significant for *$p < 0.05$; **$p < 0.01$; ***$p < 0.001$; ****$p < 0.0001$. For IF analysis 100 nuclei/samples were analyzed, the sample size was calculated on the basis of previous analysis.

**Reporting summary**. Further information on research design is available in the Nature Portfolio Reporting Summary linked to this article.

## Data availability

Uncropped/unedited gels are shown in Supplementary Fig. 16. Numerical source data are available in Supplementary data 1 and 2. All other data are available from the corresponding author on reasonable request.

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

## Acknowledgements

I would like to thank Prof. Stefano Cacchione for his generous advice and suggestions and for sharing reagents and Mr. Rocco Fraioli for technical assistance in EMSA assay. We gratefully acknowledge the Italian Association for Cancer Research for financial support [Grant 21579 to A.B.,17121 to E.S.] and Regione Lazio POR FESR Lazio 2014-2020 "Gruppi di Ricerca 2020" [Grant ID A0375-2020-36596 "ORGANOVA" to E.S. and E.V.]. E.P. and L.P. were recipients of a fellowship from the Italian Association for Cancer Research (AIRC).

## Author contributions

Conceptualization, S.E., S.A., A.A. G.G. and B.A.; methodology, validation, formal analysis, data curation, M.C., D.S.A., D.C., B.F., P.E., V.E., R.A., B.L., S.E.; investigation, D.S.A., M.C., D.C., P.E., V.E., R.A., B.A., and S.E.; writing original draft preparation, S.E.; writing review and editing, S.E., B.A., A.A., S.A., P.E., G.G.; visualization, V.E., S.E., B.A.; supervision, S.E., B.A.; project administration, S.E.; funding acquisition, S.E. and B.A.

## Competing interests

The authors declare no competing interests.
