## [Peer Review File · Communications Biology]

Reviewers' comments:

Reviewer #1 (Remarks to the Author):

In the presented manuscript, Maresca, Dello Stritto et al. describe a potential novel, direct role of PARP1 in aiding telomere replication. Mammalian telomeres have been long recognized as a notoriously difficult obstacle for DNA replication and the telomere-binding TRF1 protein was found to ensure that the replication forks progress flawlessly through telomeres (e.g. PMIDs 19596237, 19679647). The Authors now expand on these findings by showing that the PARP1 enzyme interacts with TRF1 and modifies TRF1 through PARylation, likely enabling its replication-assisting role. The Authors further ascribe this function to modification of TRF1 binding to DNA and assisting recruitment of RecQ-like helicases.

The presented data nicely show that TRF1 and PARP1 indeed interact and that PARP1 PARylates TRF1. Furthermore, PARP1 and TRF1 are found to function in the same genetic pathway to prevent telomeric DNA damage signaling and telomere fragility, a telltale of defective telomere replication. The Authors also show that TRF1 PARylation modifies two of its known core activities (PMID 7502076, 25344324) - DNA binding and interaction with the RecQ helicase BLM. They also discover a previously unappreciated interaction between TRF1 and another RecQ helicase, WRN.

Altogether, I believe the data support the core claim that PARP1 and TRF1 work together during telomere replication. This extends our understanding of how cells circumvent DNA replication errors in difficult-to-replicate loci and potentially prevent catastrophic loss of genome integrity. These are important findings that I believe should be interesting to a wider genome stability audience. That said, I find that to fully reconcile the data with a mechanistic model and to strengthen some of the Authors' claims, additional work may be warranted. I present my specific comments and suggestions for follow-up experiments below:

Major points

- 1) Even though PARylation has a very profound effect on the TRF1-DNA interaction in vitro (Fig 3D), the PARP1-dependent modulation of TRF1 binding to telomeres in cells (Fig 3C) is only very modest. Furthermore, whereas the TRF1-PARP1 interaction and TRF1 PARylation seems to peak in mid S-phase (Fig 1F, Fig 2D,E), the largest decrease in TRF1 binding to telomeres seems to happen in early S-phase (Fig 3C). How do the Authors explain these potential discrepancies? Can they perhaps comment on this in the Discussion?
- 2) The interaction between TRF1 and BLM/WRN, as well as binding of WRN to telomeres, are PARP1-dependent (Fig 3E-H). However, this does not directly prove that PARP1/TRF1 recruit these RecQ helicases to indeed facilitate telomere replication (RecQ helicases can play a multitude of roles in DNA repair). I think these data could be strengthened e.g. by performing the fragile telomere assay as in Fig 5 with a knockdown of WRN (or BLM), PARP1 and the combined knockdown of WRN (or BLM) with PARP1. If the proposed model is correct, one would expect that the knockdown of the RecQ helicases would lead to fragile telomeres, but would not further exacerbate the phenotype caused by PARP1 depletion. The genetic epistasis between WRN and TRF1 could be performed in an analogous manner, too.
- 3) It would be useful to repeat the immunoprecipitations in Fig 3E in presence of either EtBr or benzonase to exclude the possibility that the observed interactions are indirect and mediated by PARylated chromatin.

Minor points

- 4) As the quantification of fragile telomeres in Fig 5B is critical for supporting the mechanistic model, it would be beneficial to extend the analysis to three independent experiments (and >100 metaphases) instead of two as reported in the Figure legend. Of note, in the Materials and Methods section the Authors indicate that 'a total of 100 metaphases were analyzed for each sample in, at least three independent experiments' (lines 627 to 628), which seems to be contradictory to what is mentioned in

the Figure legend.

5) Can the Authors provide a quantification for the PLA control samples in Supplementary Fig 2B?

6) Perhaps I am missing something, but I do not find the data in Fig 4A particularly relevant to the rest of the manuscript. The genome-wide role of PARP1 in DNA replication has been well known (e.g. PMID 19629035, but there is a whole body of literature) and the current manuscript focuses specifically on telomeres. It may not be that necessary to show that there is a genome-wide BrdU incorporation defect, especially since a very similar experiment looking specifically at telomeres is presented in Fig 3C. I would suggest moving the data in Fig 4A to the Supplementary Info.

7) In the Discussion, the Authors mention that 'helicase recruitment [...] is necessary to remove the supercoiling generated by the fork progression' (line 467). Would helicases rather resolve DNA secondary structures instead of supercoiling (the latter being taken care of by topoisomerases)?

8) In Supplementary Fig 6E, the olaparib concentration units are indicated as millimolar. Should that maybe read micromolar? Millimolar olaparib concentrations are very much out of a pharmacologically relevant range.

Reviewer #2 (Remarks to the Author):

Maresca et al described the novel interaction between PARP1 and TRF1, which regulates the recruitment of helicases to telomeres, engaging in proper telomere replication. Utilizing both in vitro and in vivo experimental systems, the authors showed evidence that PARP1 and TRF1 directly interact in S-phase in HeLa and BJ EHLT transformed human primary fibroblasts, and PARylation of TRF1 by PARP1 disrupts its binding to telomeric DNA, which allows the recruitment of helicases, such as WRN and BLM to telomeres to assist telomere replication. Furthermore, PARP1 inhibition leads to telomere fragile phenotypes and TIF formation, and interference of TRF1 did not increase these telomere defects in PARP1 deficient cells, indicating that PARP1 acts in the same axis with TRF1 during telomere replication. The experiments were well conducted with proper controls. The manuscript was well written, but some data presentations need further clarification.

Major:

1. The authors showed that exposure of cells to replication dependent DNA damage (10mM Aphidicolin) failed to increase PARP1/TRF1 interaction, suggesting that activation of DNA damage was not involved in this process (Supplemental Figure 3 B). While this experiment design is valid, it would be of great interests to test if low dose aphidicolin (0.2 uM aphidicolin, which inhibits DNA replication polymerase alpha and slows replication fork progression, but does not cause replication fork collapse, in contrast to the high dose aphidicolin) would boost PARylation of TRF1 by PARP1 and increase TIF formations in S phase cells.

2. Which domains of TRF1 and PARP1 interact with each other? TRF1 and PARP1 interacts with many partners, which are important in telomere protection and replication. Thus, the TRF1-PARP1 interacting mutants would help dissect the function of PARP1 and TRF1 interaction from other protein-protein interactions in telomere replication.

3. Telomere replication may face great challenge in cells with long telomeres, such as ALT and Hela 1.3 (Takai, J Biol Chem. 2010 Jan 8;285(2):1457-67). Does PARP1 and TRF1 interaction plays an important role in telomere replication in cells with long telomeres.

Minor:

1. line 241: please change "wondered" to "examined".

2. Line 256: please spell the full name for EHLT.

3. Figure 3E. IP of TRF1 showed very weak BLM interaction in this figure. Is this observation repeatable? Also, in this experimental setting, is RTEL1 helicase present in TRF1 IP, since RTEL1 plays

a critical role in telomere replication?

Reviewer #3 (Remarks to the Author):

The study by Maresca et al focuses on PARP1 regulation of the shelterin component TRF1, and the consequences of disrupting this regulation. The authors claim: 1) that PARP1 PARylates TRF1 in S-phase, modifying its DNA affinity, 2) That PARP inhibition impairs telomeric replication via TRF1, and 3) That PARP inhibition prevents WRN+BLM association with the telomere, and imply that this is through TRF1.

Although I have some doubts about the novelty of the paper, the studies were well conducted.

I am convinced that the authors show

- 1) PARP1 associates with telomeres in Mid-S (Fig1)
- 2) PARP1 PARylates TRF1 strongly in Mid-S (Fig2D)
- 3) PARylated TRF1 does not bind telomeres strongly (Fig3D)
- 4) PARP1 facilitates WRN+BLM association with the telomere (Fig3E-G)
- 5) siPARP1 causes telomere replication stress and damage (Fig4;Fig5)

I would like the reviewers to address the following points

- 1) Fig1A is confusing. The text says BrdU was added 15min before the end of the second pulse, but the pictograph looks like it was added before?
- 2) Figures should be referenced in the order that they appear within the manuscript. For example, Fig1D should not be referred to before Fig1C.
- 3) Figure 1A-E should be a supplementary figure.
- 4) sFig3B-APH causes cells to become arrested in S-phase. I'm surprised, given your other results, that this did not change PARP1's interaction with TRF1/2. Could the authors elaborate beyond DNA damage?
- 5) Fig3B-C are not convincing for changes in TRF1 association with the telomere.
- 6) The authors demonstrate that increased PARylation decreases TRF1's ability to bind telomeric DNA. If this is the case, why would PARP1 to interact with more BLM/WRN when it is not bound to the telomere?
- 7) I have an issue with the Ips. They have not been treated to remove DNA (i.e. with benzonase). The interactions with TRF1 may therefore no be specific, and just represent an association with the telomere in general. The authors should confirm this by repeating some Ips with benzonase digestion before IP.
- 8) Could the authors perform the IPs in Fig3E-G with a TRF2 ab under siTRF1 conditions to show the interactions are TRF1 specific?
- 9) In Fig4D and sFig7, could the authors please provide the actual quantifications of TIFs? Only counting TIF>4 can be misleading and hid the extent of telomeric damage/replication stress.
- 10) For sFig7, the highlighted co-locs are highly dubious.
- 11) In Figure 5B you should not compare different N values between samples, they should be equal. Performing statistics between n=150 and n=69 treatments is not acceptable.

Point-by-point response to reviewers

Reviewers' comments:

Reviewer #1 (Remarks to the Author):

In the presented manuscript, Maresca, Dello Stritto et al. describe a potential novel, direct role of PARP1 in aiding telomere replication. Mammalian telomeres have been long recognized as a notoriously difficult obstacle for DNA replication and the telomere-binding TRF1 protein was found to ensure that the replication forks progress flawlessly through telomeres (e.g. PMIDs 19596237, 19679647). The Authors now expand on these findings by showing that the PARP1 enzyme interacts with TRF1 and modifies TRF1 through PARylation, likely enabling its replication-assisting role. The Authors further ascribe this function to modification of TRF1 binding to DNA and assisting recruitment of RecQ-like helicases.

The presented data nicely show that TRF1 and PARP1 indeed interact and that PARP1 PARylates TRF1. Furthermore, PARP1 and TRF1 are found to function in the same genetic pathway to prevent telomeric DNA damage signaling and telomere fragility, a telltale of defective telomere replication. The Authors also show that TRF1 PARylation modifies two of its known core activities (PMID 7502076, 25344324) - DNA binding and interaction with the RecQ helicase BLM. They also discover a previously unappreciated interaction between TRF1 and another RecQ helicase, WRN.

Altogether, I believe the data support the core claim that PARP1 and TRF1 work together during telomere replication. This extends our understanding of how cells circumvent DNA replication errors in difficult-to-replicate loci and potentially prevent catastrophic loss of genome integrity. These are important findings that I believe should be interesting to a wider genome stability audience. That said, I find that to fully reconcile the data with a mechanistic model and to strengthen some of the Authors' claims, additional work may be warranted. I present my specific comments and suggestions for follow-up experiments below:

Major points

1) Even though PARylation has a very profound effect on the TRF1-DNA interaction *in vitro* (Fig 3D), the PARP1-dependent modulation of TRF1 binding to telomeres in cells (Fig 3C) is only very modest.

We thank the reviewer for this comment. The apparent discrepancy, highlighted by this reviewer, can be explained by the fact that you are comparing an *in-vitro* with an *in-cellulo* assays. In the *in-vitro* assay, TRF1 is heavily PARylated by purified PARP1 with a very strong catalytic activity and consequently the effect is massive. In addition, in the *in-vitro* assay, recombinant TRF1 is incubated with naked DNA, which exclude the contribution of other enzymes with potential opposite effects in the process (i.e., PARG).

In *in-cellulo* assay, we detected a slight but very reproducible TRF1 detachment from chromatin, which correlates with an endogenous TRF1 PARylation (shown in Figure 2 D and E), that is more pronounced in the early S and start to be recovered in the mid-late S. In addition, although cells are synchronized, we need to consider that TRF1 is distributed on the whole telomere length, and the replication fork passage is a dynamic process. Therefore, we cannot expect to see a total detachment of TRF1 from telomeric chromatin at the endpoint analyzed.

Furthermore, whereas the TRF1-PARP1 interaction and TRF1 PARylation seems to peak in mid S-phase (Fig 1F, Fig 2D,E), the largest decrease in TRF1 binding to telomeres seems to happen in early S-phase (Fig 3C). How do the Authors explain these potential discrepancies? Can they perhaps comment on this in the Discussion?

We thank the reviewer for this comment, The new experiments allowed us to better clarify this point generating a new model that we included in the new Figure 7. We hypothesize that PARP1 PARylates TRF1 decreasing its affinity for DNA duplex in order to allow replication fork opening. Then TRF1/PARP1 complex is further stabilized by the presence of DNA, which recruits also helicases to resolve secondary structures at the G-rich strand. In agreement with this, PARP1 depletion, not only disrupts TRF1/helicase binding, but also impairs WRN recruitment at telomeric DNA in MS. All these comments have been included in the discussion section (lines 575-591).

2) The interaction between TRF1 and BLM/WRN, as well as binding of WRN to telomeres, are PARP1-dependent (Fig 3E-H). However, this does not directly prove that PARP1/TRF1 recruit these RecQ helicases to indeed facilitate telomere replication (RecQ helicases can play a multitude of roles in DNA repair). I think these data could be strengthened e.g. by performing the fragile telomere assay as in Fig 5 with a knockdown of WRN (or BLM),

PARP1 and the combined knockdown of WRN (or BLM) with PARP1. If the proposed model is correct, one would expect that the knockdown of the RecQ helicases would lead to fragile telomeres but would not further exacerbate the phenotype caused by PARP1 depletion. The genetic epistasis between WRN and TRF1 could be performed in an analogous manner, too.

We agree with the referee's point, and in the new figure 6 we interfered cells with siWRN and we measured telomere fragility in metaphase spreads. As shown, siWRN alone induces an increase in telomere fragility, as expected. More interestingly there is no significant difference between WRN interference alone and double and triple interference with siTRF1 and siPARP1 (lines 522-525).

3) It would be useful to repeat the immunoprecipitations in Fig 3E in presence of either EtBr or benzonase to exclude the possibility that the observed interactions are indirect and mediated by PARylated chromatin.

We thank the reviewer for this suggestion. We already had evidence that TRF1/PARP1 interaction was direct since the heteromodification assay *in-vitro* is performed with purified proteins and PARylation reaction requires protein-protein interaction. In addition, in the present version of the work we included a pull-down assay performed with a recombinant his-tagged wt human TRF1 incubated with a high-salt extracted Hela cell lysate, showing interaction between the two proteins. As suggested, we also repeated the experiment with EtBr (data in Supplemental Figure 9A) and we obtained very useful information that allowed us to conclude that part of the PARP1/TRF1 interactions are not mediated by DNA, while TRF1/PARP1 stabilization and helicase recruitment, especially in MS, required DNA.

Minor points

4) As the quantification of fragile telomeres in Fig 5B is critical for supporting the mechanistic model, it would be beneficial to extend the analysis to three independent experiments (and >100 metaphases) instead of two as reported in the Figure legend. Of note, in the Materials and Methods section the Authors indicate that 'a total of 100 metaphases were analyzed for each sample in, at least three independent experiments' (lines 627 to 628), which seems to be contradictory to what is mentioned in the Figure legend.

We understand the reviewer's concern and we extended the analysis to 100 metaphases for each sample as indicated in the figure legend of Figure 6 and in the materials and methods section (line 233).

5) Can the Authors provide a quantification for the PLA control samples in Supplementary Fig 2B?

We thank the reviewer for his/her suggestion, we added the quantification in the new Supplemental figure 3.

6) Perhaps I am missing something, but I do not find the data in Fig 4A particularly relevant to the rest of the manuscript. The genome-wide role of PARP1 in DNA replication has been well known (e.g. PMID 19629035, but there is a whole body of literature) and the current manuscript focuses specifically on telomeres. It may not be that necessary to show that there is a genome-wide BrdU incorporation defect, especially since a very similar experiment looking specifically at telomeres is presented in Fig 3C. I would suggest moving the data in Fig 4A to the Supplementary Info.

We agree with the referee, we moved the panel in the new Supplemental Figure 10

7) In the Discussion, the Authors mention that 'helicase recruitment [...] is necessary to remove the supercoiling generated by the fork progression' (line 467). Would helicases rather resolve DNA secondary structures instead of supercoiling (the latter being taken care of by topoisomerases)?

We agree with the referee that the RECQ helicases are specifically deputed to unwinding secondary structures, especially G-quadruplex which are abundant at telomeric G-rich sequences, more than being implicated in removing the supercoiling. We modified the discussion section accordingly (line 586-588).

8) In Supplementary Fig 6E, the olaparib concentration units are indicated as millimolar. Should that maybe read micromolar? Millimolar olaparib concentrations are very much out of a pharmacologically relevant range.

We apologize for the font mistake, of course it was micromolar concentration. It has been corrected in the legend of Supplemental Figure 8 and in the panel E.

Reviewer #2 (Remarks to the Author):

Maresca et al described the novel interaction between PARP1 and TRF1, which regulates the recruitment of helicases to telomeres, engaging in proper telomere replication. Utilizing both in vitro and in vivo experimental systems, the authors showed evidence that PARP1 and TRF1 directly interact in S-phase in HeLa and BJ EHLT transformed human primary fibroblasts, and PARylation of TRF1 by PARP1 disrupts its binding to telomeric DNA, which allows the recruitment of helicases, such as WRN and BLM to telomeres to assist telomere

replication. Furthermore, PARP1 inhibition leads to telomere fragile phenotypes and TIF formation, and interference of TRF1 did not increase these telomere defects in PARP1 deficient cells, indicating that PARP1 acts in the same axis with TRF1 during telomere replication. The experiments were well conducted with proper controls. The manuscript was well written, but some data presentations need further clarification.

Major:

1. The authors showed that exposure of cells to replication dependent DNA damage (10mM Aphidicolin) failed to increase PARP1/TRF1 interaction, suggesting that activation of DNA damage was not involved in this process (Supplemental Figure 3 B). While this experiment design is valid, it would be of great interests to test if low dose aphidicolin (0.2 μ M aphidicolin, which inhibits DNA replication polymerase alpha and slows replication fork progression, but does not cause replication fork collapse, in contrast to the high dose aphidicolin) would boost PARylation of TRF1 by PARP1 and increase TIF formations in S phase cells.

We thank the referee for the suggestion. We performed different analysis on a new experimental set upon low dose aphidicolin treatment (Figure 5) showing that 0,2 μ M aphidicolin, able to synchronize cells in S-G₂/M phases (Fig.5 A) induced an increase in TIFs positive cells at basal level and upon PARP1 interference (Fig 5 C-E). Moreover, 0,2 μ M aphidicolin treatment increased PARP1/TRF1 interaction and TRF1 PARylation (Fig. 5 B). (lines 481-491 and 600-603).

2. Which domains of TRF1 and PARP1 interact with each other? TRF1 and PARP1 interacts with many partners, which are important in telomere protection and replication. Thus, the TRF1-PARP1 interacting mutants would help dissect the function of PARP1 and TRF1 interaction from other protein-protein interactions in telomere replication.

We agree with the referee about the importance of identifying the specific domain of TRF1 involved in PARP1 interaction and hopefully the specific aminoacids which are acceptors of PARylation. Nevertheless, the only available TRF1 deletion mutant is the delta acidic one. It contains the PARylation site for Tankyrase and we used that for a PARylation in vitro assay by PARP1. The new Supplemental Figure 6 shows that this mutant is PARylated by PARP1 as well, suggesting that the interaction/parylation site of TRF1 with PARP1 is different from the Tankyrase substrate and is not in the acidic domain (lines 332-337). Unfortunately, other TRF1 deletion mutants are not available. We are currently making efforts to identify the PARP1 PARylated TRF1 amminoacids by mass spectrometry and we

hope to obtain interesting results that will be the object of a following publication.

3. Telomere replication may face great challenge in cells with long telomeres, such as ALT and Hela 1.3 (Takai, J Biol Chem. 2010 Jan 8;285(2):1457-67). Does PARP1 and TRF1 interaction plays an important role in telomere replication in cells with long telomeres.

We agree with the referee that PARP1/TRF1 interaction could play a more important role in long telomere bearing cells. To assess this point, we performed the telomere fragility experiment upon TRF1 and PARP1 interference in Hela 1.3 cells, kindly provided from Prof. T. De Lange. As shown in the Supplemental Figure 15, there is a significant increase of the occurrence of doublets/chromosome in any condition, included basal condition, but more pronounced in the case of interference of TRF1, PARP1 or both (lines 535-541 and 613-616).

Minor:

1. line 241: please change “wondered” to “examined”. We changed accordingly
2. Line 256: please spell the full name for EHLT. We included the full name
3. Figure 3E. IP of TRF1 showed very weak BLM interaction in this figure. Is this observation repeatable? The experiment has been repeated and the BLM blot was replaced.

Also, in this experimental setting, is RTEL1 helicase present in TRF1 IP, since RTEL1 plays a critical role in telomere replication? We tried to detect RTEL in the IP samples but in these condition RTEL appears not bound.

Reviewer #3 (Remarks to the Author):

The study by Maresca et al focuses on PARP1 regulation of the shelterin component TRF1, and the consequences of disrupting this regulation. The authors claim: 1) that PARP1 PARylates TRF1 in S-phase, modifying its DNA affinity, 2) That PARP inhibition impairs telomeric replication via TRF1, and 3) That PARP inhibition prevents WRN+BLM association with the telomere, and imply that this is through TRF1.

Although I have some doubts about the novelty of the paper, the studies were well conducted.

I am convinced that the authors show

- 1) PARP1 associates with telomeres in Mid-S (Fig1)

- 2) PARP1 PARylates TRF1 strongly in Mid-S (Fig2D)
- 3) PARylated TRF1 does not bind telomeres strongly (Fig3D)
- 4) PARP1 facilitates WRN+BLM association with the telomere (Fig3E-G)
- 5) siPARP1 causes telomere replication stress and damage (Fig4;Fig5)

I would like the reviewers to address the following points

- 1) Fig1A is confusing. The text says BrdU was added 15min before the end of the second pulse, but the pictograph looks like it was added before?

We thank the reviewer for having noticed that and we apologize for the mistake, the correct schedule of treatment was the one illustrated in the pictograph. We corrected the text accordingly.

- 2) Figures should be referenced in the order that they appear within the manuscript. For example, Fig1D should not be referred to before Fig1C.

We thank the referee and we corrected the citation order within the text.

- 3) Figure 1A-E should be a supplementary figure.

We agree with the referee, and we moved the cell cycle analysis (Figure 1 B) in the Supplemental Figure A.

- 4) sFig3B-APH causes cells to become arrested in S-phase. I'm surprised, given your other results, that this did not change PARP1's interaction with TRF1/2. Could the authors elaborate beyond DNA damage?

We thank the reviewer for the comment. High dose aphidicolin treatment, induces massive replication dependent DNA damage, and do not affect TRF1/2 interaction with PARP1. Instead, low dose aphidicolin treatment induces replication stress and S-phase accumulation allowing us to study the role of TRF1/PARP1 axe in the resolution of replication problems at telomeres. For this reason, we performed additional experiments upon low dose of aphidicolin treatments. This type of treatment was able to induce a replication stress able to block cells in S-phase, increase TRF1/PARP1 interaction, TRF1 PARylation and TIFs induction both at basal level and, with a higher extent, upon TRF1/PARP1 interference (Figure 5) (lines 481-491 and 600-603).

- 5) Fig3B-C are not convincing for changes in TRF1 association with the telomere.

We thank the reviewer for this comment. In *in-cellulo* assay we detected a slight but very reproducible TRF1 detachment from chromatin, which correlates with an endogenous TRF1 PARylation (shown in Figure 2 D and E), is more pronounced in the early S and starts to be recovered in the mid-late S. Moreover, although cells are synchronized, we need to consider

that TRF1 is distributed on the whole telomere length, and the replication fork passage is a dynamic process, then we cannot expect to see a total detachment of TRF1 from telomeric chromatin at the endpoint analyzed. On the contrary, in the *in-vitro* assay, where TRF1 is heavily PARylated by purified PARP1 with a very strong catalytic activity we observe a massive decrease in TRF1/DNA affinity.

6) The authors demonstrate that increased PARylation decreases TRF1's ability to bind telomeric DNA. If this is the case, why would PARP1 to interact with more BLM/WRN when it is not bound to the telomere?

We thank the reviewer for this comment. Thanks to the additional experiments performed we had the opportunity to clarify this point. We hypothesize that PARP1 PARylates TRF1 decreasing its affinity for DNA duplex, which allows replication fork opening. Successively, TRF1 binding to telomeric DNA is rapidly recovered and TRF1/PARP1 complex is further stabilized by the presence of DNA, as shown in the new Supplemental Figure 9, and recruits helicases putatively to resolve secondary structures on the G-rich single strand. In agreement with this, PARP1 depletion, not only disrupts TRF1/helicase binding, but also impairs WRN recruitment at telomeric DNA in MS. All these comments have been included in the discussion section (lines 575-591).

7) I have an issue with the Ips. They have not been treated to remove DNA (i.e., with benzonase). The interactions with TRF1 may therefore not be specific, and just represent an association with the telomere in general. The authors should confirm this by repeating some Ips with benzonase digestion before IP.

We understand the reviewer concern and, as above indicated, we repeated the experiments with EtBr to remove DNA from the complex obtaining further clarification of the dynamics of complex formation. We have evidence that there is a direct association between TRF1 and PARP1 also from the heteromodification assay *in-vitro*, which is performed with purified proteins, and requires protein-protein interaction. In addition, in the present version of the work we included a pull-down assay performed with a recombinant his-tagged wt human TRF1 incubated with a high-salt extracted Hela cell lysate, showing interaction between the two proteins (Supplemental Figure 9 B, lines 397-404 and 576-591).

8) Could the authors perform the IPs in Fig3E-G with a TRF2 ab under siTRF1 conditions to show the interactions are TRF1 specific? As suggested by the referee we performed IPs with an anti-TRF2 ab under siTRF1 condition and we analyzed association of TRF2 with PARP1, WRN and BLM.

We thank the reviewer for the suggestion. As shown in the Supplemental Figure 10, all the three proteins are bound by TRF2. Moreover, while WRN and BLM binding seems to be not affected by cell cycle progression or by TRF1 interference, PARP1 binding increases in MS-LS/G2 phases (as previously shown in Supplemental Figure 4A) and the increase is reduced by siTRF1. This is very interesting since it confirms the prevalent role of TRF1/PARP1 interaction in telomere replication compared to TRF2 (lines 405-412 and 571-576).

9) In Fig4D and sFig7, could the authors please provide the actual quantifications of TIFs? Only counting TIF>4 can be misleading and hid the extent of telomeric damage/replication stress.

We thank the referee for the suggestion. The number of TIFs/cell has been scored and added where requested.

10) For sFig7, the highlighted co-locs are highly dubious.

We improved the quality and the resolution of the images

11) In Figure 5B you should not compare different N values between samples, they should be equal. Performing statistics between n=150 and n=69 treatments is not acceptable.

We fully agree with the referee's concerns. To address this point, we performed additional experiments to reach a total number of 100 metaphases scored in each sample.

REVIEWERS' COMMENTS:

Reviewer #1 (Remarks to the Author):

I have reviewed the revised version of the manuscript and find that the Authors have addressed my previous concerns. I have no further comments to add.

Reviewer #2 (Remarks to the Author):

The authors have addressed my questions.

Reviewer #3 (Remarks to the Author):

The Authors have adequately addressed all my previous concerns. I recommend this manuscript for publication.